# An Imidazolium-Based Ionic Liquid as a Model to Study Plasticization Effects on Cationic Polymethacrylate Films

**DOI:** 10.3390/polym15051239

**Published:** 2023-02-28

**Authors:** Thashree Marimuthu, Zainul Sidat, Pradeep Kumar, Yahya E. Choonara

**Affiliations:** Wits Advanced Drug Delivery Platform Research Unit, Department of Pharmacy and Pharmacology, School of Therapeutic Sciences, Faculty of Health Sciences, 7 York Road, Parktown, Johannesburg 2193, South Africa

**Keywords:** ionic liquids, green chemistry, plasticization, polymeric films, mechanical properties, stress/strain properties

## Abstract

Ionic liquids (ILs) have been touted as effective and environmentally friendly agents, which has driven their application in the biomedical field. The study compares the effectiveness of an IL agent, 1-hexyl-3-methyl imidazolium chloride ([HMIM]Cl), to current industry standards for plasticizing a methacrylate polymer. Industrial standards glycerol, dioctyl phthalate (DOP) and the combination of [HMIM]Cl with a standard plasticizer was also evaluated. Plasticized samples were evaluated for stress–strain, long-term degradation, thermophysical characterizations, and molecular vibrational changes within the structure, and molecular mechanics simulations were performed. Physico-mechanical studies showed that [HMIM]Cl was a comparatively good plasticizer than current standards reaching effectiveness at 20–30% *w*/*w*, whereas plasticizing of standards such as glycerol was still inferior to [HMIM]Cl even at concentrations up to 50% *w*/*w*. Degradation studies show HMIM-polymer combinations remained plasticized for longer than other test samples, >14 days, compared to glycerol <5 days, while remaining more pliable. The combination of [HMIM]Cl-DOP was effective at concentrations >30% *w*/*w*, demonstrating remarkable plasticizing capability and long-term stability. ILs used as singular agents or in tandem with other standards provided equivalent or better plasticizing activity than the comparative free standards.

## 1. Introduction

Over the years, the structural properties of ionic liquids have been leveraged for the design of advanced materials that have been applied to the development of diverse technologies and fields [1]. Film-forming materials derived from synthetic or natural polymers can serve as protective barriers [2] or functional materials [3] coupled with tuneable mechanical properties for applications in various fields. Nonetheless, the development of film-forming materials, for example, is challenged by poor mechanical properties and relative brittleness, which often necessitates the incorporation of plasticizers [4]. Plasticizers are generally molecular compounds that are small in nature and when introduced into the matrix of the polymer, may result in the lowering of the glass transition temperature (Tg) of said polymers. This creates flexible but strong materials that can withstand the large stresses often associated with production and processing methods [5]. However, plasticizer effects are not limited to Tg and can alter the degree of crystallinity, electric conductance and optical clarity, amongst other physical properties [6].

The most frequently used plasticizers on the market are phthalate-based ([4]), for example, dioctyl phthalate (DOP), which have been in widespread use since the 1930s [7]. However, this particular class of plasticizers has garnered attention in recent years due to concerns over their relative ubiquity, slow biodegradation, leaching and safety. In addition, they are suspected of negative health impacts, including endocrine side effects and anti-androgenic propensity [6,8,9].

Other common plasticizers include polyols such as glycerol and sorbitol, which are especially effective in plasticizing hydrophilic polymers [10]. However, glycerol has limited activity when evaluated with hydrophobic polymers, making the resulting agent highly hygroscopic [11], which is not ideal for applications that may require a moisture-free environment.

The multitude of issues plaguing current plasticizers, such as poor safety, long biodegradation times, leaching and limited activity [12], have driven the need for more effective, environmentally friendly, biologically safe, biodegradable plasticizers. In the early 2000s, ionic liquids (ILs) have historically been touted as “greener” safer agents, and now they remain as designer solvents with high tunability with regard to the selected anions and cations that constitute their chemical makeup [13]. The applications of ILs have been recently reviewed by Alfonso et al. [13] and Kaur et al. [14], where solvent-aided separation and extraction, dissolution, chemical synthesis, plasticization, electrochemical, and analytical methods, environmental protection and advanced materials were highlighted. 

Today, there are evidenced-based discussions where the greenness of ILs is being re-examined [13,15]. However, they remain interesting plasticizer candidates for advanced materials applications such as cellulose electronic substrates [16], cellulosic materials as flame retardants [17], starch melts [18] and PVA fibers with better mechanical properties [19].

Furthermore, polymeric-based pressure sensors were developed from carrageenan with IL concentrations up to 40% *w*/*w* of 1-butyl-3-methylimidazolium tetrachloroferrate ([Bmim][FeCl_4_] [20]. Another interesting study demonstrated that ILs could be incorporated into porous metal–organic frameworks (MOFs). The resulting interactions facilitated the movement of cations, which enabled the system to serve as solid electrolytes [21]. 

In another study on solid polymeric electrolytes, Barbosa et al. developed microporous silicate-based and imidazolium IL systems, thus further highlighting the versatile applications of ILs in the design of advanced materials [22].

ILs based on imidazolium or pyridinium cations bear a strong structural resemblance to conventional plasticizers [9], with many ILs being great plasticizer candidates as they exhibit several of the desirable properties such as low volatility and leachability, high thermal and chemical stability and solvation compatibility with a wide range of polymers to list a few [23]. 

Incidentally, investigations on the thermal properties of ILs are ongoing to gain a deeper understanding of the structure–property relationships. For example, Ferdeghini et al. coupled experimental and computational approaches to observe the thermal behavior of N-morpholinium dicationic ionic liquids [24]. Similarly, the nature of the cations has been shown to influence the thermal stability of ILs; and alternative ILs like alkylphosphonium-based ILs have demonstrated higher thermal stability [25]. Moreover, an alternative and relatively more efficient synthesis for a model IL, namely, 1-butyl-3-methylimidazolium chloride ([Bmim]Cl), has been reported by Chiappe and colleagues [26]. Such methods could lower cost and environmental impact, encouraging further studies with imidazolium-based ILs. 

A number of authors have recognized the applicability of ILs as plasticizers for polymethacrylates [27,28,29]. Eudragit RL100 is a polymethacrylate derivative that has shown widespread application in the development of pharmaceutical formulations with the controlled release [30]. This study evaluates the use of multiple commonly used plasticizers from different classes against a relatively non-toxic at specified concentrations—and monocationic IL within the Eudragit RL100 polymer. The agents explored in this study (Figure 1) include phthalates such as DOP, polyols such as glycerol and ionic liquids such as 1-Hexyl-3-methylimidazolium chloride ([HMIM]Cl). 

The IL was chosen due to its comparatively environmentally friendly footprint and extremely low toxicity (ranked 162 out of 232 for AChE toxicity rank) [31]. The toxicity and biodegradability concerns of imidazolium-based ionic liquids are also correlated to their applications and environmental aspects [32]. Although the model IL; 1-hexyl-3-methylimidazolium chloride [HMIM][Cl] has been previously applied as a plasticizer for pharmaceutical film formulations for topical drug delivery, its application was limited due to dose-dependent toxicity concerns [33]. Alternative ILs such as L-carnitine C6 alkyl ester bromide (Carn6) and betaine C6 alkyl ester bromide (Bet6) have been reported and evaluated at an in vitro and in vivo level as a topical formulation for ocular administration [34]. Similarly, a novel IL was designed by Tampucci and co-workers for topical drug delivery that utilized the combination of tetramethylguanidinium cation with naturally derived hydrophobic fatty acid carboxylates [35]. 

Nonetheless, this study utilized a readily available IL that served as a structural model for the plasticization of Eudragit RL100 polymer. Therefore, the focus of this study is to explore the physicochemical properties, electrostatic interactions, comparative long-term stability of the resulting produced films, and corresponding stress/strain properties acquired through a model IL and conventional plasticizers. 

## 2. Materials and Methods

All chemicals and solvents procured were of analytical grade. Materials comprising of 1-hexyl 3-methyl imidazolium chloride (202.72 g/mol, ≥97.0% (HPLC)), dioctyl phthalate (390.56 g/mol, ≥99.5%), glycerol (92.09 g/mol, ≥99.5%), and all solvents were procured from Sigma-Aldrich (St. Louis, MO, USA). Eudragit RL100 polymer (pellet form, (average 32,000 g/mol) was procured from Evonik.

### 2.1. Representative Method for Preparation of the Control Film and Plasticized Films

The polymer solution was prepared using solution-casting techniques. In line with green chemistry principles [36,37], a co-solvent mixture was employed in proportions that optimized performance and decreased the need for volatile chemical solvents. An amount of 100 g of Eudragit-RL100 pellets was dissolved in a co-solvent mixture composed of 170 mL acetone, 160 mL isopropyl alcohol and 10 mL water. The resulting solution was modified with the relevant plasticizer and poured using a funnel mounted at a fixed height (50 cm) onto a silicone base mat which facilitated easy sample extraction. Contamination was limited using a fume hood set at a constant airflow (30 m/s) with temperature control (25 °C) and controlled solvent evaporation achieved by inverting the funnel over the sample area to prevent rapid solvent loss. Films were easily removed and required only slight pressure at one edge from a tweezer. Several films were prepared and are illustrated in Table 1. Prior to characterization, all films were stored at room temperature and relative humidity of 25 °C and 60%, respectively.

### 2.2. Determination of Thermophysical Properties of the Fabricated Films

Thermophysical properties of plasticized films were determined using differential scanning calorimetry (DSC) (Mettler Toledo, DSC, STARe System, Schwerzenbach, ZH, Switzerland). Plasticized film and control samples of 8–10 mg were prepared and sealed in aluminium crucibles which were then run in a temperature range of 18–400 °C at an incremental heating rate of 10 °C/min under a nitrogen gas purged atmosphere (100 mL/min). Data were collated and plotted as heat flow against temperature. 

### 2.3. Determination of Molecular Vibrational Transitions of the Fabricated Films

FTIR spectra of the fabricated films were analyzed for specific interactions between the pristine polymer (Eudragit-RL100) and plasticizing agents. The spectra allowed for the determination of plasticization effect, theoretical maximum concentrations, efficacy, and mechanism of plasticizing agents. A PerkinElmer Spectrum 2000 ATR-FTIR (PerkinElmer 100, Llantrisant, Wales, UK) was used to evaluate samples excised to uniform sizes appropriate for the sample area and loaded onto a spectrometer fitted with a single-reflection diamond MIRTGS detector. FTIR was reported with a wavenumber range of 4000–650 cm^−1^. Detailed FTIR for pristine polymer film and formulations at 10% *w*/*w* of plasticizer can be found in the Appendix A.

### 2.4. Determination of Long-Term Stability of Fabricated Films

Samples were evaluated for stability and plasticity over a 28-day cycle. Appropriately sized samples were excised and placed in an airtight container and placed in an agitated chamber with controlled temperature (37 °C) and humidity in an orbital shaker incubator LASEC LM-530 (Yihder, Taiwan). Samples were weighed before being placed in the container and then again at regular intervals to determine weight loss. 

### 2.5. Determination of Stress/Strain Properties of Fabricated Films

Uniaxial tensile strain testing was conducted on the fabricated plasticized films using a BioTester 5000 Biomaterials Tester (CellScale, Waterloo, ON, Canada). Tests were conducted using a 5 N load cell with an accuracy rating of 5 mN under ambient conditions (25 °C) to assess the displacement, rigidity, strength, and plasticity effect. Prior to sample mounting, the BioTester was calibrated and balanced, and arms placed at a distance measuring 950 mm. Samples were prepared by excising a square of ~2 cm × ~2 cm from the entire film with the aid of a custom cutting implement provided by CellScale. Samples were then mounted using a pair of BioRakes^TM^ composed of multiple tines (thin tungsten wires), each of which pierced the samples, anchoring them at opposing edges (Figure 2) with no preload applied to the sample. The sample was then subjected to a controlled-displacement test; as such, the stress levels varied from sample to sample. The samples were expected to withstand a 20% ramp displacement deformation stress and were then evaluated for tearing, peak size, peak force, peak displacement, and stress distribution patterns. Data were collated, and stress–strain values were obtained and used to calculate Young’s modulus. 

### 2.6. Multiple-Point Thickness Test of the Fabricated Films

Thickness of the fabricated plasticized films was evaluated using a digital caliper. The digital caliper was calibrated and zeroed prior to use each time. Using three points equidistant from each point, the film was measured and recorded with an average of the 3 readings calculated and reported in this study. 

### 2.7. Folding Endurance of Fabricated Plasticized Films

A section of approximately 25–30% of the total film surface area produced was excised and used for testing. The excised film section was folded at a central portion and pressed flat. This was repeated until such point as a break, crack or extreme deformation was noted. To ensure validity, accuracy and reliability, the same pestle and motions were replicated from one test to the next. Using the pestle ensured blunt rather than penetrating trauma to the fabricated films.

### 2.8. Mechanistic Profiling of IL-EUD Complex via Molecular Mechanics Simulations

Molecular mechanics simulations were carried out (HyperChem^TM^ 8.0.8 Molecular Modelling Software) between the ionic liquid and the polymer at 1:1 molecular ratio, wherein the chemical structures were generated employing natural bond angles. The complex structure was developed by parallel disposition of constituent molecules, and an assisted model building and energy refinements force field was applied (in vacuo; Polak–Ribiere Conjugate Gradient) for energy minimization.

## 3. Results and Discussion

### 3.1. Evaluation of Molecular Vibrational Transitions of the Fabricated Films

The FTIR spectrum of the polymer and plasticized films were recorded to determine if there were any interactions between the plasticizers and polymer (Figure 3a–d). All plasticized film FTIR spectra depicted characteristic peaks of Eudragit RL 100 at ~3500 cm^−1^ (OH stretching), ~2950 cm^−1^ (CH stretch) and ~1750 cm^−1^ (CO stretch), which is comparable to the literature values [38]. Notably, the position of these molecular vibrations showed no significant changes relative to the spectrum of the pristine film.

DOP plasticized films showed a slight increase in intensity for the band at 3350 cm^−1^ (OH stretch, Figure 3a, encircled A), a sharp medium peak increase at 2960 cm^−1^, (alkane CH stretch Figure 3a, encircled B), a weak sharp change at the 1250 band, (CO stretch, Figure 3a, encircled C) and associated small peaks with slight shifts and intensity variations at 1140, 1730 (CO stretch) and 1450 (CH alkane bend) cm^−1^.

In contrast, glycerol plasticized films showed strong, broad peaks at 3500 cm^−1^ (OH stretch, (Figure 3b, encircled A) and corresponding CO stretching at ~1035 cm^−1^ (Figure 3b, encircled B). The intensity of both bands increased as the concentration of glycerol increased, which could also be indicative of weak polymer–plasticizer interactions. The excessive brittleness noticed in the testing of glycerol-based samples can be perhaps attributed to the abovementioned change to the polymeric chains, causing instability once the plasticizer migrates. 

[HMIM]Cl plasticized samples show increasing intensity at 3500 cm^−1^ (OH stretch Figure 3c, encircled A), which can be attributed to the possible interaction between water and acid protons on the imidazolium groups at C2 and the O atoms of the ester groups in the polymethacrylate polymer. Similar findings were reported for ionic liquids based on imidazolium and ammonium when applied as a plasticizer for cellulosic films [17]. According to Ramenskaya, a slight shift of Δ10 for the C = O stretching band was indicative of weak C2 –H⋯O = C hydrogen interactions for films of poly(methyl methacrylate) and 1-butyl-3-methylimidazolium hexafluorophosphate (BMIPF_6_) [39]. In this study, there were no significant changes observed for the C = O stretching band (Figure 3c) for all formulations. However, there are additional peaks at 2960 cm^−1^ (CH alkane stretch Figure 3c, encircled b), a weak sharp peak at ~1572 cm^−1^ (alkene, C = C stretching Figure 3c, encircled C), a medium sharp peak at 950 and a weak peak at 820 cm^−1^ (CH bending). The presence of HMIM is indicated by new bands at ~1572 cm^−1^, attributed to C = C alkene vibrations.

[HMIM]Cl -DOP blend samples displayed broad peaks at ~ 3500 cm^−1^ (OH stretch, Figure 3d, encircled A), medium sharp peaks at 2950 and 2850 cm^−1^ (CH alkane stretching, (Figure 3d, encircled B), a weak sharp peak at 1572 cm^−1^ (C = C stretching, cyclic alkene, Figure 3d, encircled C) and a strong, sharp peak at 1250 cm^−1^ (CO stretching). In the blend plasticized samples, the FTIR spectra show additive changes verifying the presence of the two plasticizers that have been combined.

### 3.2. Determination of Thermophysical Properties of the Fabricated Films

DSC thermograms of the control and plasticized films are presented in Figure 4a–d. Relative to the thermogram of Eudragit RL film (control), DOP plasticized films (Figure 4a) show minor changes. However, a noticeable change in the thermograms around the ~250–290 °C range was observed as the % *w*/*w* of [HMIM]Cl increased in the films (Figure 4b), which can potentially be attributed to the loss of IL from the polymeric film [40].

The thermograms of blend samples (Figure 4c) follow the trend observed in the FTIR spectra where additive changes are noted, including the 100 °C range similar to DOP and the 200–300 °C range as seen with [HMIM]Cl-based samples. In Figure 4d, glycerol samples show little to no change at lower concentrations but sharp changes in the 250–300 °C at concentrations exceeding 30%. Due to the amorphous nature of the polymer, various small variations from the baseline are observed throughout all DSC thermograms (Figure 4a–d), which can be largely attributed to minor changes in the conformational structure from the addition of plasticizer.

### 3.3. Determination of Long-Term Stability, Thickness and Endurance of Fabricated Films

Long-term stability, endurance and thickness tests were conducted and are summarized in Table 2. Long-term stability studies of the samples show a clear trend with an increased loss of mass peaking around 5–10% *w*/*w* plasticizer concentration, followed by a sharp but gradual decline reaching the lowest values at 50%. However, glycerol-plasticized samples do not follow this trend, with their behavior difficult to correlate based on concentration. Glycerol samples have the highest mass loss at 1% *w*/*w* but continue to lose mass sporadically and erratically over the study period. The thickness of the synthesized films was, in most cases, lower when compared to the control, with a few notable exceptions. Some DOP samples (2%, 30% and 50%) and all glycerol-based samples had higher average thickness than the control. For DOP samples, all differences fall into the region of ±0.05 mm compared to the control. Glycerol-based samples do not follow the same observation with differences up to 0.17 mm thicker than the control. 

The control was able to withstand >200 folds before breaking during endurance testing, which differs by up to 70 folds in other studies [41,42] and is attributed to differences in synthetic methods, including the use of different solvents and the generation of films using a petri dish with borders which limits how wide or large the film may be. In this study, no impeding walls were used, allowing the fluid to flow freely and settle, which impacted both thickness (decreased by ~0.1 mm) and endurance rating (decreased by ≤70 folds). Endurance ratings of the different plasticized films follow a simple pattern with a direct correlation between concentration and endurance, with concentrations >10% *w*/*w* seemingly inadequate. Endurance ratings (from best to worst) follow the trend blend > [HMIM]Cl > DOP > glycerol.

### 3.4. Determination of Stress/Strain Properties in the Fabricated Films

All tested samples except glycerol were able to withstand the maximum force applied (Figure 5). The distribution of forces along the samples was not uniform, and deformation in some cases was irreparable (Figure 6). Stress, strain, and Young’s modulus values have been calculated and are reported in Figure 7 and Figure 8, respectively.

The best plasticizing activity is obtained using the blend that demonstrated remarkable activity with concentrations as low as 1% *w*/*w*, yielding a 3 times improvement in Young’s modulus. Plasticizing effectiveness can thus be summarized as blend > DOP > [HMIM]Cl > glycerol, which is similar to the trend observed in other physical tests, such as stability studies. However, this test also provides information on deformation characteristics and how stress and strain are spread across the entire system. Deformation patterns in the control group and glycerol samples show poor stress distribution over the entire system compared to [HMIM]Cl samples. [HMIM]Cl shows distribution patterns that ensure the entire system averages strain and the deformation has a trapezoid-like formation with slightly higher strains at the points where stress is initially applied. DOP samples show better strain distribution patterns when compared to the control but are still sub-optimal with clear hotspots, while blend plasticized samples fall between DOP and [HMIM]Cl with better distribution of stresses than DOP, but not as effective as [HMIM]Cl-based samples. These are perhaps due to weak interactions between the [HMIM]Cl hydrogen moiety of the imidazolium ring (N=CH−N) and the carbonyl ester groups on the polymer interacting [43]. Notably, a Cl^−^ anion may also play a role in plasticizing the polymer but results in weaker interactions as opposed to other anions for polymer-IL interactions such as those found in other works where [HMIM]PF_6_^−^ plasticized PMMA was more significant [44].

The molecular simulation of the IL-EUD molecule complex confirmed very high energy stabilization patterns characterized by all bonding and non-bonding energy contributions (except for the electrostatic interactions) (Table 3). The total energy of the complex was even less than that of the polymer molecule itself. This confirmed not only the compatibility and incorporation of IL within the polymer matrix but also the close geometrical proximity of the molecules in the form of sliding molecular structures (Figure 9). A destabilized electrostatic component is actually essential for the movement of the plasticizer between the polymer chains.

## 4. Conclusions

The use of [HMIM]Cl as a plasticizer as either a singular agent or an additive with another plasticizer is an effective technique to improve the mechanical properties of compatible polymers. These agents can be used as advanced materials applications at low concentrations (~5%) to achieve plasticizer actions, limiting the amount needed and helping to alleviate concerns of toxicity and accumulation. Compared to traditional phthalate plasticizers such as DOP, [HMIM]Cl, as used in this study, was proven to be as effective when used as a singular agent and provided additive effects when used in tandem. [HMIM]Cl proved to be a worthy alternative and provided plasticizer effects at concentrations that were 6× lower when compared to glycerol. The plasticized films produced in this study were stable for relatively long periods and showed good distribution of mechanical stresses when tested. Furthermore, molecular mechanics simulations showed that the model IL when applied as a plasticizer, can act as a filler within the polymeric matrix. This is particularly relevant when incorporating ILs within porous materials. Using the model IL, future work is warranted for the use of biocompatible ILs that can be synthesized through sustainable methods to ensure the holistic implementation of green principles.

## Figures and Tables

**Figure 1 polymers-15-01239-f001:**
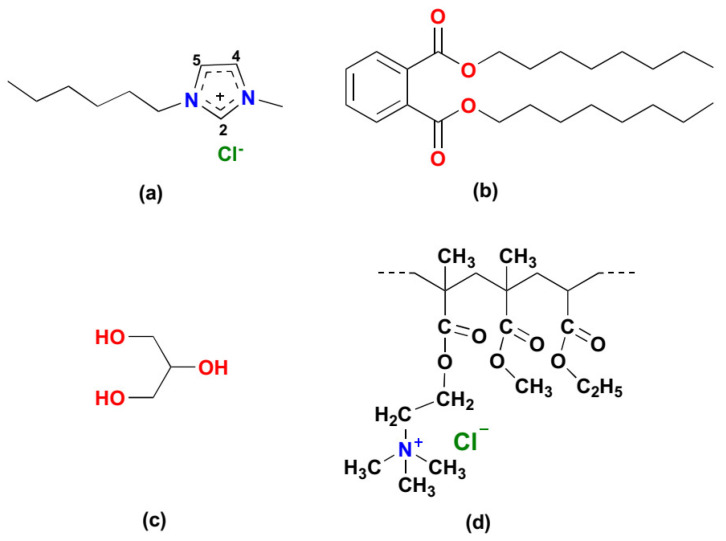
Structure of plasticizers: (**a**) 1-hexyl 3-methyl imidazolium chloride ([HMIM]Cl); (**b**) dioctyl phthalate (DOP); (**c**) glycerol and polymer: (**d**) Eudragit RL 100 or Poly(ethyl acrylate-co-methyl methacrylate-co-trimethylammonioethyl methacrylate chloride.

**Figure 2 polymers-15-01239-f002:**
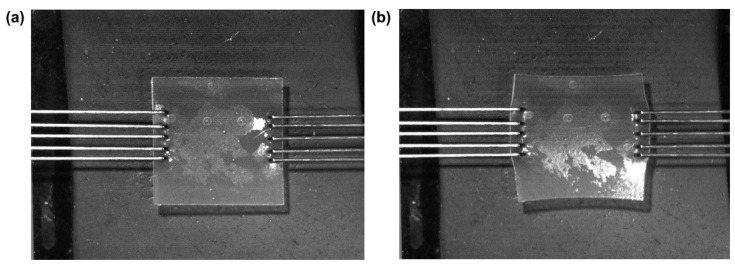
Illustration of sample setup within the CellScale BioTester. (**a**) The setup before any stress is applied; (**b**) at the end of the test after stress has been applied and deformation can clearly be seen.

**Figure 3 polymers-15-01239-f003:**
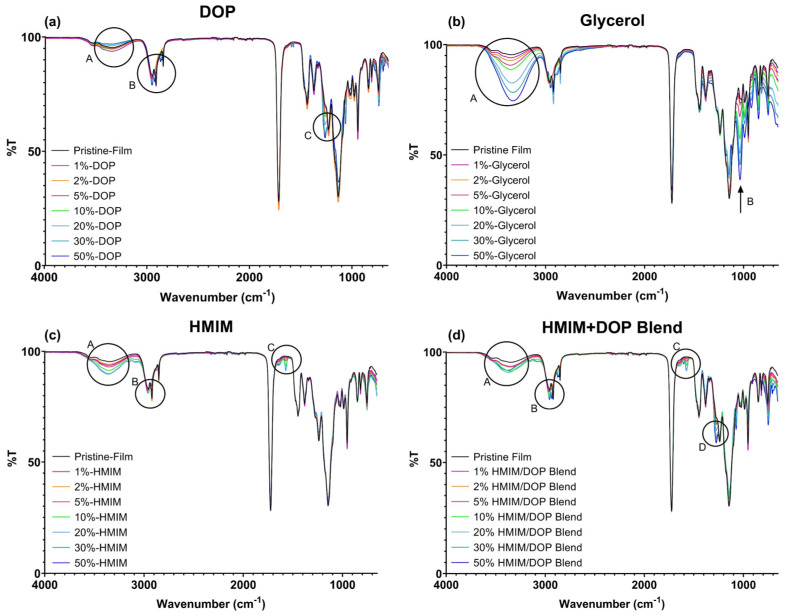
FTIR Spectra of (**a**) DOP, (**b**) Glycerol, (**c**) HMIM and (**d**) HMIM and DOP blend plasticized samples with varying concentrations.

**Figure 4 polymers-15-01239-f004:**
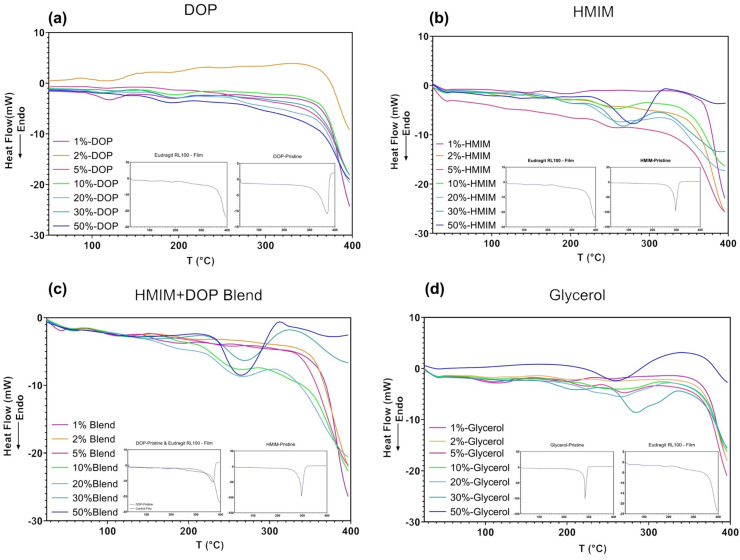
DSC thermogram of (**a**) DOP, (**b**) HMIM and (**c**) HMIM and DOP blend (**d**) Glycerol plasticized samples. (insert contain reference thermograms).

**Figure 5 polymers-15-01239-f005:**
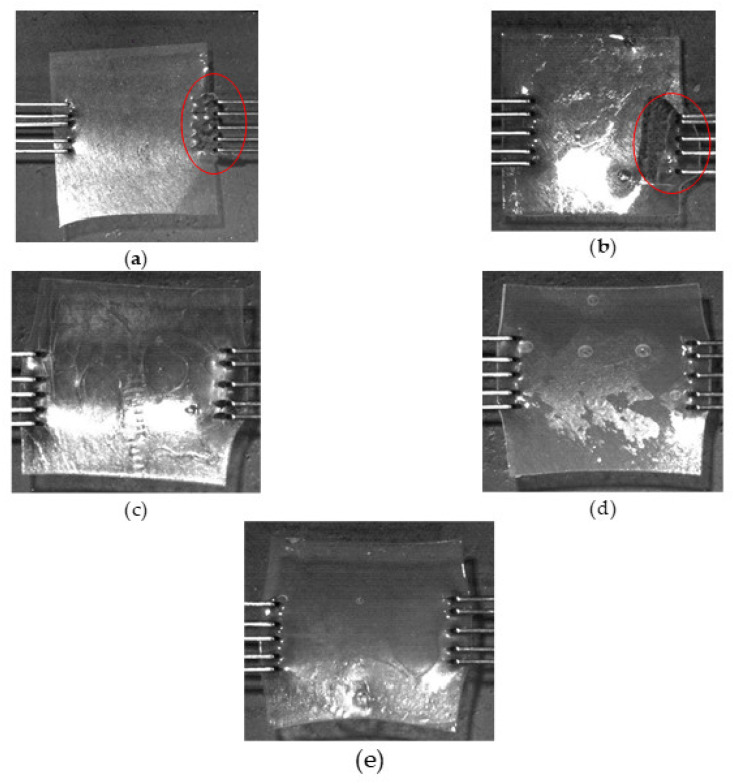
Images showing the end outcome of the stress–strain tests. End of test images of (**a**) 50% *w*/*w* glycerol (ruptured); (**b**) Eudragit control film (ruptured); (**c**) 50% *w*/*w* DOP; (**d**) 50% *w*/*w* [HMIM]Cl; (**e**) 50% *w*/*w* blend.

**Figure 6 polymers-15-01239-f006:**
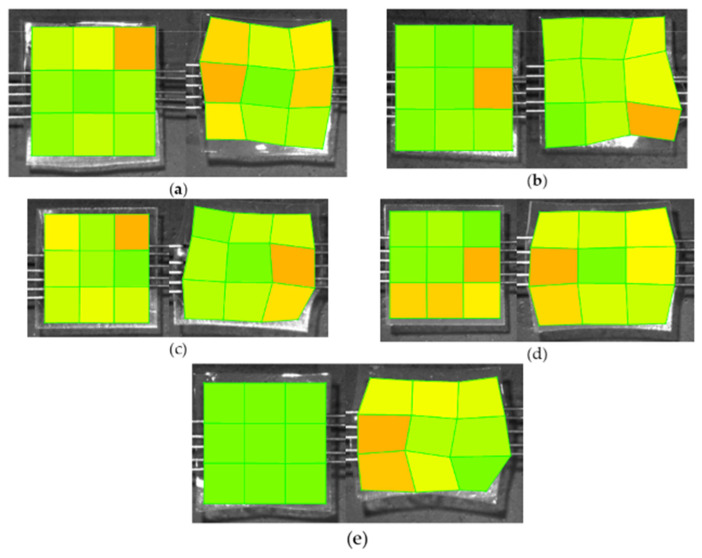
Images show the strain patterns (low strain—green; high strain—orange) before any applied stress (Left) and after the stress test has completed (Right). (**a**) 50% *w*/*w* glycerol (ruptured); (**b**) Eudragit control film (ruptured); (**c**) 50% *w*/*w* DOP; (**d**) 50% *w*/*w* [HMIM]Cl; (**e**) 50% *w*/*w* blend. Images with source points were generated using the LabJoy software interface accompanying the BioTester equipment.

**Figure 7 polymers-15-01239-f007:**
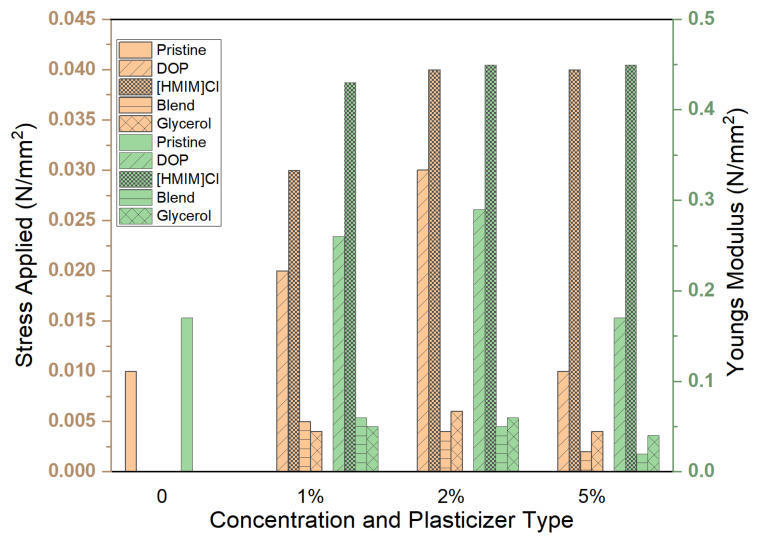
Effect of plasticizer type and concentration 1, 2 and 5% on the endured stresses and strains as a ramp displacement-type stress test was conducted. The resultant Young’s modulus has been calculated. For all samples, the strain endured was 0.1.

**Figure 8 polymers-15-01239-f008:**
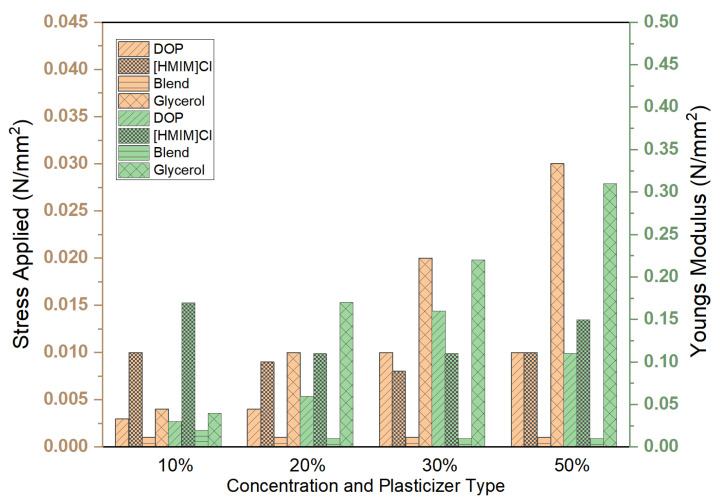
Effect of plasticizer type and concentration 10, 20, 30 and 50% on the endured stresses and strains as a ramp-displacement-type stress test was conducted. The resultant Young’s modulus has been calculated. For all samples, the strain endured was 0.1.

**Figure 9 polymers-15-01239-f009:**
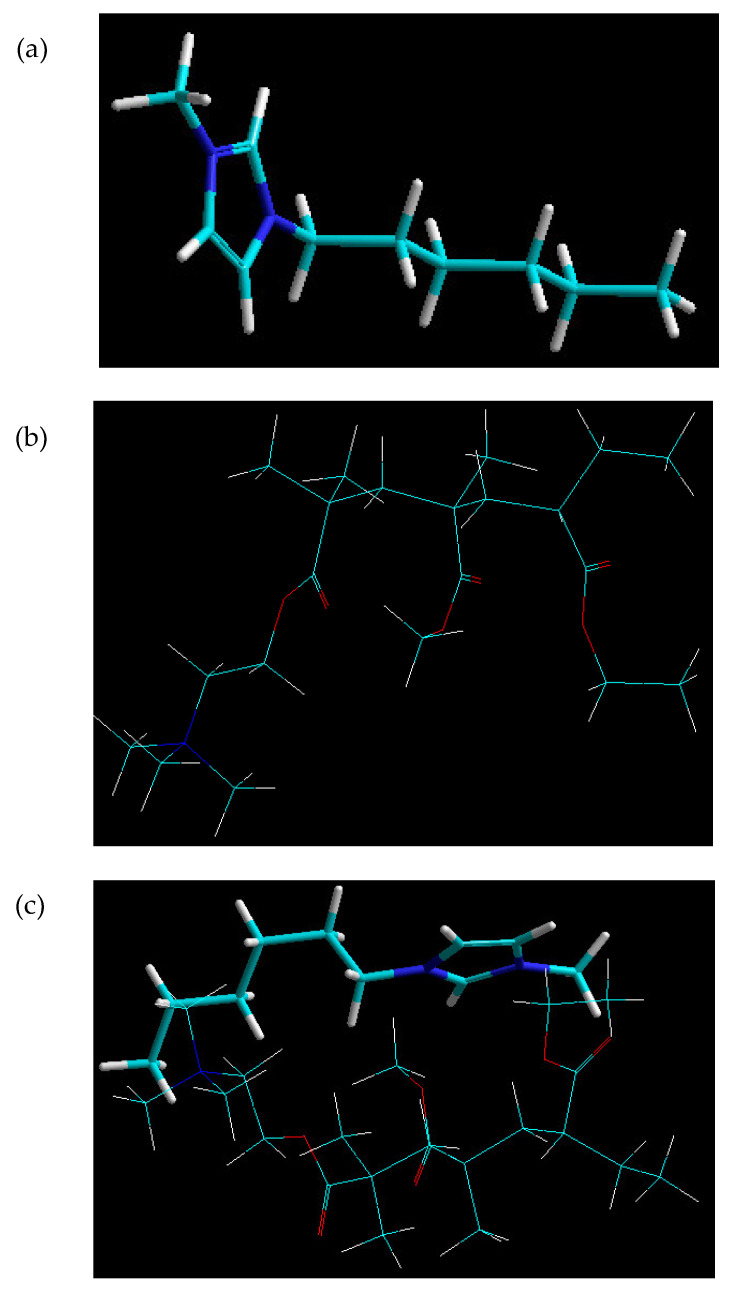
Minimum energy confirmation for (**a**) IL, (**b**) EUD and (**c**) molecular mapping and geometrical positioning of IL and EUD after molecular mechanics simulations in a vacuum.

**Table 1 polymers-15-01239-t001:** Samples prepared and analyzed.

Sample	Plasticizer ^1^
Pristine-Film (PF)	-
[HMIM]Cl-RL100	[HMIM]Cl
DOP-RL100	DOP
GLY-RL100	Glycerol
Blend-RL100	[HMIM]Cl & DOP

^1^ Percentages calculated using dry polymer weight amounts (*n* = 3) Several samples per type of plasticizer were prepared where the amount of plasticizer was 1%, 2%, 5%, 10%, 20%, 30%, and 50% *w*/*w*, respectively.

**Table 2 polymers-15-01239-t002:** Summary of physical tests is outlined in the table below, including total mass lost from samples over the testing period, thickness, and endurance rating.

Plasticizer	Concentration [% *w*/*w*] ^1^	Stability (Loss of Mass in mg)	Average Thickness (mm)	Endurance (# of Maximum Folds)
Pristine	0%	32.7	0.37	205
DOP	1%	6.4	0.29	200
	2%	19.7	0.41	198
	5%	10.8	0.28	207
	10%	6.2	0.24	236
	20%	5.0	0.36	239
	30%	15.0	0.39	212
	50%	5.3	0.42	215
[HMIM]Cl	1%	12.6	0.26	195
	2%	20.4	0.26	213
	5%	24.3	0.25	226
	10%	34.1	0.24	265
	20%	26.2	0.22	269
	30%	11.6	0.26	218
	50%	12.2	0.24	232
Blend	1%	8.6	0.30	199
	2%	12.5	0.28	242
	5%	54.6	0.32	245
	10%	40.4	0.28	251
	20%	7.6	0.25	248
	30%	19.6	0.22	231
	50%	2.2	0.29	238
Glycerol	1%	8.6	0.30	199
	2%	12.5	0.28	242
	5%	54.6	0.32	245
	10%	15.6	0.47	220
	20%	7.6	0.44	229
	30%	12.4	0.54	234
	50%	2.5	0.48	234

^1^ Percentages were calculated using dry polymer weight amounts.

**Table 3 polymers-15-01239-t003:** Molecular attributes for the IL-polymer in silico complex in vacuo.

Molecular Attributes ^1,2^	IL	Eudragit	IL-EUD	ΔE
Total energy	17.496	38.216	33.380	−22.332
Bond energy	0.175	3.834	2.773	−1.236
Angle energy	16.095	17.034	23.653	−9.476
Dihedral angle energy	1.203	11.749	12.752	−0.2
van der Waals energy	0.023	5.598	−10.417	−16.038
Electrostatic energy	0.0	0.0	4.619	+4.619

^1^ All energy values are in kcal/mol. ^2^ ΔE = energy difference between the constituent molecules and the complex.

## Data Availability

The data emanating from this study are available on request from the corresponding author.

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
