# Peer review of "An Imidazolium-Based Ionic Liquid as a Model to Study Plasticization Effects on Cationic Polymethacrylate Films"

_polymers, 2023, doi:10.3390/polym15051239_

Round 1

Reviewer 1 Report

This paper reported the plasticizing effect of an imidazolium-based IL and compared with other two traditional additives DOP and glycerol. Plasticized samples were evaluated for stress-strain, long-term degradation, thermophysical characterizations and molecular vibrational changes within the structure. It showed that the IL was a superior plasticizer in an appropriate concentration. In the main text, there are some points to be improved:

(1) In the experimental part, the details of the materials used in this study is lost. The information, such as purity, Mw and etc. should be included.

(2) How was the Figure 6 plotted/calculated? by software or by hand? 

(3) In table 3, the unit of Youngs Modulus is lost.

(4) In conclusion, "the rheological" properties was not discussed in this study.

(5) Overall, it is an interesting work to investigate the effect of IL in polymer matrix, however, the "superior" of the used IL said by the authors is not very clear in the analysis process.

Author Response

Thank you for the valuable comments.

Reviewer 2 Report

The paper titled "An Imidazolium-Based Ionic Liquid as a Green Alternative for Plasticization of Cationic Polymethacrylate Films" describes the use of IL as a plasticizer in methacrylate polymer. The authors compare IL rispero to other standard plasticizers such as glycerol, Eudargit and dioctyl phthalate (DOP). The work is well structured and the results are clearly shown. The use of ILs with plasticizers has been extensively reproduced. In particular, the study is reminiscent, perhaps too much so, of what was reported in Reference 15. The authors should better explain the increase to knowledge in this work compared to the literature such as not using fluorinated ILs. There are also other benefits to using ILs in polymeric materials (10.1021/acssuschemeng.2c02374, 10.1002/adem.202200849, 10.3389/fchem.2022.995063). This aspect should be further explored in the introduction. The part on ILs in the introduction itself also has some gaps and critical issues. First of all, it is not clear why the authors talk specifically about the use of ionic liquids as plasticizers in pharmaceutical applications and in transdermal in particular.  The selected ionic liquid can hardly be deployed in this field given the toxicity of imidazole ionic liquids (10.1016/j.jhazmat.2007.10.079). Other ILs can be used in famraceutics but have lower toxicity and are of natural origin (10.3390/ijms24032714, 10.3390/pharmaceutics12111078). This aspect should be clarified and eventually these considerations should be added. In addition, the part on applications (line 57) and properties (line 63) should absolutely be added with relevant literature (Application: 10.1039/D2GC03198A, 10.1021/acs.oprd.6b00302 Properties: 10.1016/j.molliq.2021.115662, 10.1016/j.molliq.2021.115892).

For ease of reading, Table 3 should be transformed into a graph (e.g., histogram).

Only after these revisions and proper motivation for the selection of IL the paper can considered suitable for publication.

Author Response

Thank you for the expert opinion which has improve the manuscript.

Round 2

Reviewer 2 Report

I would like to thank the authors for accepting my suggestions and for the clarity on the rationale of the study performed. Considering the overall changes made, I recommend the paper for publication.